# Mixed methods evaluation of the Getting it Right First Time programme in elective orthopaedic surgery in England: an analysis from the National Joint Registry and Hospital Episode Statistics

Helen Barratt ,[1] Andrew Hutchings ,[2] Elena Pizzo ,[1] Fiona Aspinal ,[1] Sarah Jasim ,[3] Rafael Gafoor ,[1] Jean Ledger ,[1] Raj Mehta ,[1] James Mason ,[4] Peter Martin ,[1] Naomi J Fulop ,[1] Stephen Morris ,[5] Rosalind Raine [1]

HB and AH are joint first authors.

For numbered affiliations see end of article.

**Correspondence to**
Dr Andrew Hutchings;
andrew.hutchings@lshtm.ac.uk

## ABSTRACT

**Objective** To evaluate the impact of the 'Getting it Right First Time' (GIRFT) national improvement programme in orthopaedics, which started in 2012.

**Design** Mixed-methods study comprising statistical analysis of linked national datasets (National Joint Registry; Hospital Episode Statistics; Patient-Reported Outcomes); economic analysis and qualitative case studies in six National Health Service (NHS) Trusts.

**Setting** NHS elective orthopaedic surgery in England.

**Participants** 736 088 patients who underwent primary hip or knee replacement at 126 NHS Trusts between 1 April 2009 and 31 March 2018, plus 50 NHS staff.

**Intervention** Improvement bundle including 'deep dive' visits by senior clinician to NHS Trusts, informed by bespoke set of routine performance data, to discuss how improvements could be made locally.

**Main outcome measures** Number of procedures conducted by low volume surgeons; use of uncemented hip implants in patients >65; arthroscopy in year prior to knee replacement; hospital length of stay; emergency readmissions within 30 days; revision surgery within 1 year; health-related quality of life and functional status.

**Results** National trends demonstrated substantial improvements beginning prior to GIRFT. Between 2012 and 2018, there were reductions in procedures by low volume surgeons (ORs (95% CI) hips 0.58 (0.53 to 0.63), knees 0.77 (0.72 to 0.83)); uncemented hip prostheses in >65 s (OR 0.56 (0.51 to 0.61)); knee arthroscopies before surgery (OR 0.48 (0.41 to 0.56)) and mean length of stay (hips −0.90 (−1.00 to -0.81), knees −0.74 days (−0.82 to −0.66)). The additional impact of visits was mixed and comprised an overall economic saving of £431 848 between 2012 and 2018, but this was offset by the costs of the visits. Staff reported that GIRFT's influence ranged from procurement changes to improved regional collaboration.

**Conclusion** Nationally, we found substantial improvements in care, but the specific contribution of GIRFT cannot be reliably estimated due to other concurrent

## STRENGTHS AND LIMITATIONS OF THIS STUDY

⇒ We report the first, independent evaluation of the high profile Getting It Right First Time (GIRFT) National Health Service improvement programme.

⇒ Our mixed-method approach enabled us to provide a comprehensive and robust understanding of GIRFT, exploring the impact of the programme from different perspectives.

⇒ Our linked dataset allowed us to examine a range of measures, as well as estimating the specific contribution of Trust visits, whilst the case study analysis provided further insights from the perspective of Trust staff.

⇒ We could not examine some key GIRFT target measures (eg, procurement, litigation rates) because appropriate data were not available.

⇒ We also could not capture costs incurred by Trusts, because activities were not consistently tracked.

initiatives. Our approach enabled additional analysis of the discrete impact of GIRFT visits.

## INTRODUCTION

'Getting it Right First Time' (GIRFT) is one of the largest improvement programmes in the National Health Service (NHS). It began in orthopaedics in 2012 with the publication of the first GIRFT report, recommending changes to improve outcomes and reduce costs.[1] Following government investment totalling £62.8m, GIRFT now operates in 44 different specialties or clinical workstreams.[2] GIRFT is an improvement 'bundle'—a small number of interventions performed together to improve care.[3] Clinical leadership is fundamental: GIRFT was established by a senior surgeon, who leads the orthopaedic

workstream and now chairs the wider programme. Each workstream is chaired by a clinical lead from the relevant specialty. The programme includes components operating at local (ie, NHS provider Trusts) and national (ie, across England) levels. Local components include 'deep dive' visits to Trusts, while national components include national reports describing how unwarranted variations can be addressed. Initial first visits were piloted at a small number of Trusts in 2013, then replicated across the country in orthopaedics and other workstreams from 2015. Before each visit, Trusts are sent a bespoke 'data-pack,' collated by GIRFT, describing their performance on over 100 variables, drawn from sources including Hospital Episode Statistics (HES) and national audits, such as the National Joint Registry (NJR). Data include use of evidence-based procedures and costs. Data packs bring data sources together and make them accessible, facilitating comparison to national and peer group averages. Discussion at the meeting is driven by the datapack and tailored to the Trust. Attendees, comprising clinicians, managers and other relevant professionals, identify where and how improvements could be made.Revisits follow a similar format with Trusts reporting changes made since the first visit.

There are few examples of initiatives on this scale, and consequently few evaluations. The American College of Surgeons National Surgical Quality Improvement Programme (NSQIP) provides data to drive improvement. However, only around 10% of US hospitalsparticipate.[4] In the UK, the national Perioperative Quality Improvement Programme will collate and feedback data about surgical complications.[5] NHS Trusts are also subject to inspections by the Care Quality Commission, while some National Clinical Audits also provide written feedback.[6] However, GIRFT clinical leads visit all Trusts to discuss theirdatapack,as well asopportunities for improvement. GIRFT reflects evidence that measurement alone is insufficient.[7] The effectiveness of feedback is dependent on hard' and 'soft'[8] incentives, including using data to support change and holding participants to account at revisits.[9] However, although GIRFT visits Trusts twice, sometimes more, the largest improvements in NSQIP were seen where providers had participated for many years.[10] Improvements were also more marked at siteswith pre-existing internal improvement programmes.[11] GIRFT makes use of system (top-down) leadership, but designatedlocal (bottom-up) leadership within Trusts may also be important.[12] Despite the limited evidence base, GIRFT has been expanded across the NHS, in the absence of independent evaluation.

We, therefore, evaluated the effectiveness and cost-effectiveness of the GIRFT orthopaedic workstream, focusing on the most common elective procedures, primary hip and knee replacement.[13] The few evaluations of national programmes focus mainly on quantitative process and outcome measures, and hence may miss wider effects. Consequently, we undertook a multicomponent analysis. The first described national trends in processes

and outcomes over time, starting before GIRFT. However, this cannot disentangle the relative impact of GIRFT from other concurrent national initiatives that targeted similar measures. For example, Lord Carter's review of NHS hospital efficiency took place between June 2014 and February 2016, and made recommendations about tackling 'unwarranted variation' in key resource areas such as staffing, diagnostics and procurement. NHS Right Care originated as part of the Quality, Innovation, Productivity and Prevention programme in 2009 and is still ongoing. It supports local economies to improve population healthcare and address performance variations. Our second analysis focused on a specific component of GIRFT, first visits to Trusts, to elucidate their additional local impact over and above the national trends. We also undertook an economic analysis to evaluate the cost impacts of visits, and a qualitative exploration of the impact of GIRFT from the perspective of Trust staff.

## METHODS
### Design
We employed a mixed-methods approach, using both quantitative and qualitative data to consider the various impacts of GIRFT from different perspectives and at different levels (national and Trust). In the quantitative analysis, we examined eight key GIRFT target measures. We first describe changes in the seatnational level. We then exploit variation in the timing of GIRFT visits to assess the *additional* impact of initiating local involvement after allowing for national trends. We also undertook an economic analysis to assess costs and savings attributable to the visits. Finally, we used qualitative methods to explore how staff perceived that GIRFT had impacted practice at six case study sites.

### Quantitative methods
#### Quantitative data sources
We used linked data from NJR, HES and patient-reported outcome measures (PROMs) to identify patients aged 18 or over whounderwent an elective primary hip or knee replacement between 1 April 2009 and 31 March 2018, at 126 English Trusts thatreceived a first GIRFT visit by November 2017. We excluded eight Trusts visited later because they would contribute no postvisit data to the analysis. We included primary procedures eligible for the PROMs programme.[14] Metal-on-metal implants were excluded because they were subject to specific regulatory measures.[15]

#### Variables
We identified eight measures that were GIRFT priorities and available from the dataset. We were unable to use data on some priority indicators, such as litigation and procurement, because they are not publicly available. Process measures were: (1) procedures conducted by low volume surgeons ≤35 similar procedures or ≤10 unicondylar/patella-femoral knee replacements per

year); (2) use of uncemented hip implants in patients >65; (3) arthroscopy in year prior to knee replacement. GIRFT sought to reduce these, citing evidence that higher surgeon volumes equate to better outcomes; knee arthroscopy is not effective; and uncemented hip implants in older patients have higher revision rates.[1] There is no commonly accepted threshold for surgeon volume. We therefore used the thresholds ≤35 similar procedures or ≤10 unicondylar/ patella-femoral knee replacements per year based on clinical input, the literature on 'low volume' surgical thresholds in orthopaedic surgery, discussion with the GIRFT programme team and recommendations in their 2015 programme report.[2] Outcome measures comprised: (1) hospital length of stay for index procedure; (2) emergency readmissions within 30 days; (3) revision surgery within 1 year; and in the subset of patients who participated in the national PROMs programme (4) health-related quality of life (EQ-5D)[16] and (5) functional status (Oxford Hip/ Knee Score (OHKS)).[17] GIRFT sought to minimise the first three to reduce associated costs. We included EQ5D and OHKS to facilitate economic evaluation. We identified revisions from NJR and HES,[18] measuring patient-reported outcomes at 6 months.[14] We adjusted for: age in years; ethnicity (white, not white, or missing/not reported); sex; quintile of Index of Multiple Deprivation; Charlson comorbidity[19] (HES) and American Society of Anesthesiologists grade (NJR).

## Statistical analyses

We examined proportions and means of measures before GIRFT started (2009/2010–2011/2012), and in the final 2 years of the study (2016/2017–2017/2018). In our analysis of national trends, we also estimated year-on-year changes in comparison with 2012/13, the year GIRFT began, using casemix-adjusted hierarchical logistic and linear regression.

We analysed the additional impact of initiating visits using a pre–post design. First visit dates defined preperiods and postperiods for Trusts. The first visit marked the start of GIRFT's local involvement with an individual Trust. Prior to this, a Trust would be aware of GIRFT's national work, but would not have received the intervention tailored to their individual circumstances. At the start of the programme, the GIRFT team contacted all eligible Trusts, and the timetable of the first visits was based on the order in which sites responded, rather than, for example, orthopaedic performance data. Nevertheless, the order of visits was not random, so we divided Trusts into early, middle and late groups in a 1:2:1 ratio, to examine whether changes differed by visit timing. The ratio was specified a priori to split Trusts by lower and upper quartiles, ensuring a minimum of 30 per group. We used hierarchical logistic and linear regression models adjusted for casemix variables with clustering at Trust level to estimate changes in levels for each measure. We controlled for temporal trends using fractional polynomials, and preoperative scores in PROMs analyses. The impact in early, middle and late groups were estimated using interactions

between group and the change in levels. The only casemix variables with missing data were ethnicity (3.8% and 2.7% of THR and TKR patients, respectively) and Index of Multiple Deprivation (table 1). We used a complete-case analysis but included a missing category for ethnicity. Cases with a missing deprivation score represented only 0.3% of the analysis sample and were excluded from the complete-case analysis. We allowed for implementation delays by excluding procedures in the 3 months after each initial Trust visit, based on a GIRFT case study indicating that improvements occurred within 3 months.[20] We used a longer 15-month exclusion when analysing arthroscopies, because these were measured in the 12 months before surgery and hence might overlap the visit date.

## Economic analysis methods

Conventionally, economic evaluation compares costs and outcomes of an intervention and comparator. However, it was not possible to use a comparator because GIRFT visited Trusts at different times. Instead, we compared the impact of visits at Trust level with expected costs in the absence of GIRFT. We examine: (1) cost of the visits; (2) costs incurred by Trusts to implement recommendations; and (3) costs or savings resulting from the visits in the limited measures publicly available for analysis (figure 1).

Information from the GIRFT programme team enabled us to calculate visit costs at Trust level. We collected data on costs incurred by Trusts via a national survey, distributed electronically on our behalf, in early 2018, by the programme team, to GIRFT 'champions' in each Trust. It included questions about five GIRFT recommendations that could have an economic impact: implementation of ring-fenced beds; introduction of extended physiotherapy services; changes in use of theatre loan kits; reductions in activity by low volume surgeons; and improvements to theatre efficiency.

We assessed the economic impact of changes using the results of the Trust-level analysis above, quantified using NHS cost data.[21] We used the marginal effects estimated in the statistical models to estimate the economic impact of the visits where a statistically significant change was observed.

## Qualitative methods

As outlined in the study protocol,[13] our evaluation included several qualitative elements, including interviews with the GIRFT programme team and national health leaders (to understand the development of GIRFT and its evolution over time), as well as focus groups with patients (to understand their views about the content of the GIRFT programme). As these elements did directly contribute to our assessment of the impact of GIRFT, their findings will be reported elsewhere. We used a case study approach to explore the implementation of GIRFT in individual NHS Trusts, including staff perceptions about whether and how GIRFT impacted practice locally.[22]

**Table 1** Characteristics of patient population*

| | Hip replacement patients (n=337 279) | | Knee replacement patients (n=398 809) | |
|---|---|---|---|---|
| | N | % | N | % |
| **Age** | | | | |
| 10–19 | 37 | 0.0 | – | – |
| 20–29 | 1553 | 0.5 | 140 | 0.0 |
| 30–39 | 4805 | 1.4 | 851 | 0.2 |
| 40–49 | 16 765 | 5.0 | 10 443 | 2.6 |
| 50–59 | 46 753 | 13.9 | 53 736 | 13.5 |
| 60–69 | 96 371 | 28.6 | 131 197 | 32.9 |
| 70–79 | 115 840 | 34.4 | 144 271 | 36.2 |
| 80–89 | 51 900 | 15.4 | 55 873 | 14.0 |
| 90 and over | 3255 | 1.0 | 2298 | 0.6 |
| **Sex** | | | | |
| Male | 136 468 | 40.5 | 169 959 | 42.6 |
| Female | 200 811 | 59.5 | 228 850 | 57.4 |
| **Ethnicity** | | | | |
| White | 317 671 | 94.4 | 360 742 | 90.5 |
| Other | 6941 | 2.2 | 27 327 | 6.9 |
| Missing | 12 667 | 3.8 | 10 740 | 2.7 |
| **Index of Multiple Deprivation (IMD) quintile** | | | | |
| Least deprived | 78 059 | 23.1 | 84 401 | 21.2 |
| Less deprived | 80 505 | 23.9 | 88 714 | 22.2 |
| Deprived | 61 831 | 18.3 | 77 281 | 19.4 |
| More deprived | 59 340 | 17.6 | 75 173 | 18.9 |
| Most deprived | 56 520 | 16.8 | 72 153 | 18.1 |
| Missing* | 1024 | 0.3 | 1087 | 0.3 |
| **Charlson Comorbidity Score** | | | | |
| 0 | 210 385 | 62.4 | 232 953 | 58.4 |
| ≥1 | 126 894 | 37.6 | 165 856 | 41.6 |
| **American Society of Anesthesiologists Score** | | | | |
| Healthy | 40 379 | 12.0 | 32 329 | 8.1 |
| Mild systemic | 227 471 | 67.4 | 284 064 | 71.2 |
| Severe/moribund | 69 429 | 20.6 | 82 416 | 20.7 |

*Cases missing an IMD score were excluded from the complete-case analysis.
ASA, American Society of Anesthesiologists Score; IMD, Index of Multiple Deprivation.

## Case study sites

We purposively sampled six Trusts in England, representing a spread of hospital types (district general and teaching hospitals) and setting (region and rural vs urban).

## Qualitative data collection

We have described our data collection methods previously.[13] Here, we report data collected via semistructured interviews with staff at the six case study sites, between October 2016 and May 2019 (online supplemental appendix 1). The interview topic guide was developed with input from the evaluation team (eg, to incorporate questions about resource costs and implementation) and informed by scoping discussions conducted with the team delivering GIRFT, to understand the programme components (see online supplemental appendix 2). It was piloted prior to the interviews, and then refined iteratively as the study progressed, to take account of emerging findings. Interviewees included surgeons and other staff present at the first GIRFT visit or knowledgeable about local improvement. Interviews lasted between 20 and 60 min and were audiorecorded forfull transcription.

| | **What** | **How** | **Data source** |
|---|---|---|---|
| **Cost of GIRFT visits** | Cost of GIRFT team<br>Data pack production<br>Cost of visits trips<br>Cost of visits time<br>Cost of Trusts staff time | Staff wages<br>Cost of data pack<br>Mileage cost x total miles<br>Transport/meeting time x wage/hr<br>Meetings time x wage/hr | NHS, PSSRU<br>GIRFT team data |
| **Cost for Trusts to implement recommendations** | Ring fenced<br>Extended physiotherapy<br>Theatre load kits<br>Minimum Volumes<br>Theatre Efficiency | National Survey to NHS Trusts | National Survey to NHS Trusts in August 2018 |
| **Economic impact of changes in clinical practice and outcomes** | Arthroscopy procedures<br>Fixation methods<br>Readmissions<br>Revisions<br>Length of Stay<br>PROMS-QALYs | Quantitative analysis results<br>Volumes x costs of procedures | NJR and HES<br>National Tariffs<br>NICE reports<br>HRGs |

**Figure 1**  The economic components of GIRFT orthopaedic evaluation. GIRFT, Getting it Right First Time; HES, Hospital Episode Statistics; HRGs, Healthcare Resource Groups; NHS, National Health Service; NICE, National Institute for Health and Care Excellence; PROMs, patient-reported outcomes measures; QALYs, quality-adjusted life-years.

### Qualitative data analysis

We analysed data thematically, within and across cases, combining inductive and deductive (informed by the aims of the GIRFT programme) approaches.[23]

### Patient and public involvement

From the inception of the study, we worked with the NIHR CLAHRC North Thames lay Research Advisory Panel to to refine the protocol and ensure that the proposed research appropriately reflected the priorities, experience, and preferences of patients. Through this, we identified a patient representative (RM) who agreed to join the study steering board. He has subsequently played a collaborative role in refining the research design, interpreting findings and disseminating results. The research reported here did not directly involve patients, so RM did not play a role in recruitment to the study.

## RESULTS
### Quantitative findings
#### Study population

A total of 337 279 patients who underwent a hip replacement and 398 809 who received a knee replacementin 126 NHS Trusts, between 1 April 2009 and 31 March 2018 (figures 2 and 3) (see table 1 for patient characteristics).

#### National trends
*Process measures*

Nationally, there were substantial improvements in process measures,often beginning before 2012 when GIRFT (figures 4 and 5). Comparing 2009–2012 and 2015–2018, reductions varied from 29% fewer uncemented hips to a 58% reduction in knee arthroscopy (table 2). Reductions in procedures by low volume surgeons began prior to GIRFT for hips and knees. OR for 2017/2018 vs 2012/2013 were 0.58 (95% CI 0.53 to 0.63) for hips and 0.77 (95% CI 0.72 to 0.83) for knees. Reductions in uncemented hips and knee arthroscopy also began prior to

2012, but the largest occurred as GIRFT progressed with ORs for 2017/2018 vs 2012/2013 of 0.56 (95% CI 0.51 to 0.61) and 0.48 (95% CI 0.41 to 0.56), respectively.

*Outcome measures*

Mean length of stay reduced by just over 1 day for both hip and knee patients between 2009–2012 and 2015–2018. The largest reductions began before GIRFT (figures 4 and 5). Mean differences between 2012/2013 and 2017/2018 were −0.90 days (95% CI −1.00 to −0.81) for hips and −0.77 days (95% CI −0.82 to −0.66) for knees. There was some evidence that 1 year knee revisions declined by 2017/18, but little evidence of a change in hips. Postoperative quality of life and functional status improved, but there was little change in emergency readmissions (table 2 and figures 4 and 5).

#### Additional effect of initiating 'deep dive' involvement at individual trusts

GIRFT's first visits to Trusts occurred between September 2013 and November 2017 with half occurring between February and July 2014 (table 3).

*Process measures*

The additional effects of the visits on process measures, after controlling for national trends, differed between the earliest Trusts visited and the middle and later groups. In the earliest group, we observedreductions in procedures by low volume surgeons. Conversely, this increased in the middle and late groups, but use of uncemented hipsreduced (table 3).

*Outcome measures*

Following visits, mean length of stay increased for the middle and late groups, but decreased in hip patients in the earlier group. However, estimated effects were small in comparison with national trends. We found limited evidence of an impact on other outcomes (table 3).

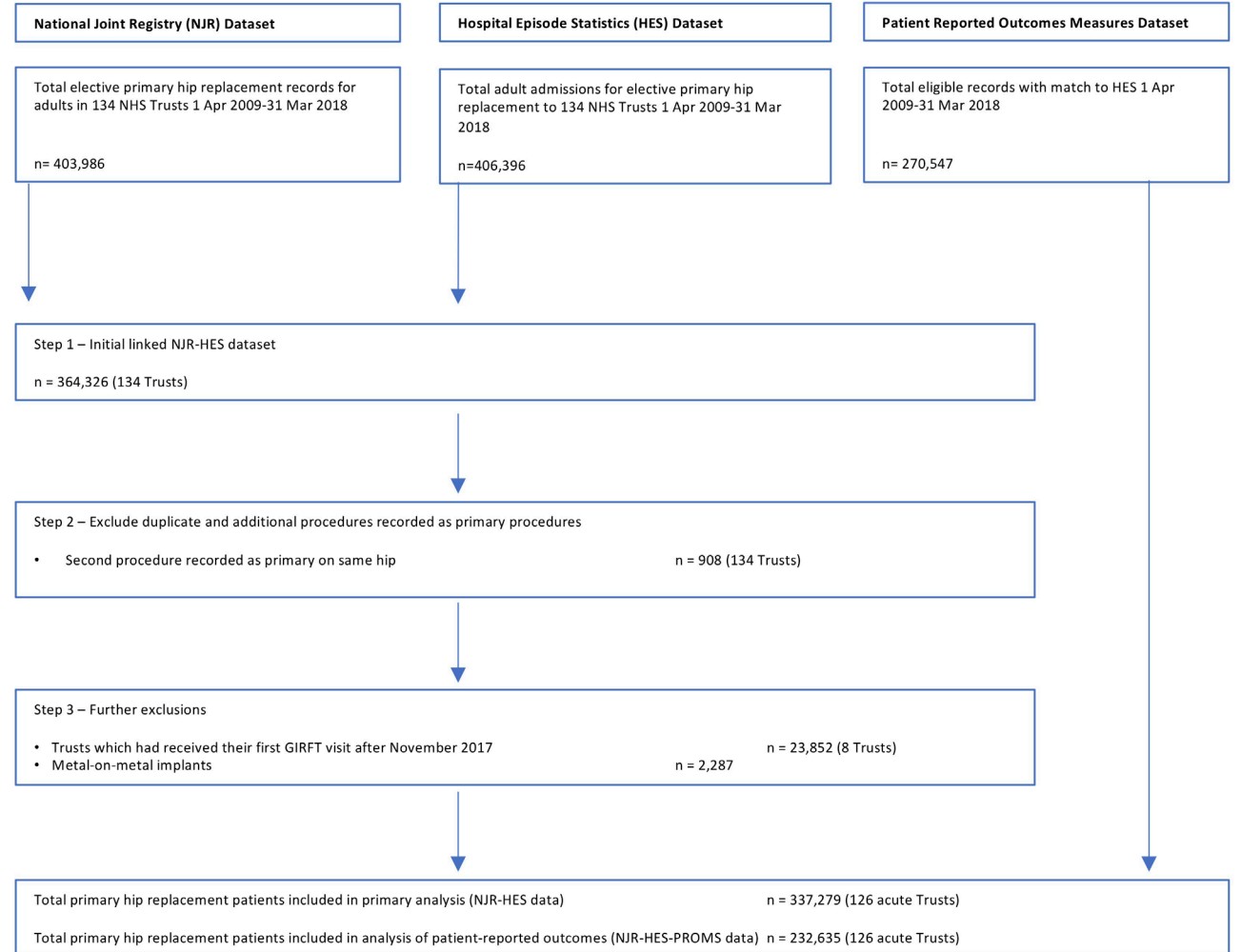

**Figure 2** Study population—total hip replacement patients. Patient-reported outcomes measures (PROMs) data collected from April 2009 for consenting patients only. GIRFT, Getting it Right First Time; NHS, National Health Service.

## Economic analysis

The estimated cost to deliver the Trust visits was £491 420 (component 1). This is a 'sunk' cost, including £10 769 transport costs (total return trip mileage); £150 000 for datapacks; £111,916 GIRFT staff time; and £106 820 opportunity cost of Trust staff time attending the meetings.

Our national survey provided mixed reports about the implementation of recommendations so it was not possible to accurately estimate costs. Therefore, although Trusts incurred costs, these could not be included (component 2).

Although length of stay increased after the visits (p<0.05), this was not included in our economic analysis. Trusts receive the same payment for each inpatient stay, up to the tariff 'trim point' (currently 9 days). However, average length of stay remained below this, despite the observed increase, so there was no extra monetary cost. Nevertheless, increased length of stay does come with an opportunity cost, as the bed is not available for another patient. As there was no change in EQ-5D, used to calculate quality-adjusted life-years, we were unable to undertake a cost-effectiveness analysis. Instead, we conducted a cost–consequences analysis. We observed positive and negative changes after GIRFT visits (table 3) and additional costs partly outweighed savings (table 4). The overall impact of the statistically significant changes, for the limited number of measurable variables, not including length of stay, was a saving of £431 848 (component 3).

## Case-study analysis

In this paper, our goal is to evaluate the effectiveness and cost-effectiveness of the GIRFT programme, focusing on processes and outcomes of care. We; therefore, restrict our reporting here to the qualitative data sources where the impact of GIRFT on patient care was explored: interviews conducted at our six case study sites. We conducted 50 interviews across six sites (online supplemental appendix 1). Interviewees described five types of impact, operating at three levels: individual, Trust and regional (figure 6). These ranged from changes to implant selection, to improved networking within Trusts and across regions. However, GIRFT particularly impacted ways of working at Trust level, for example, catalysing planned improvements.

Interaction within and between the three levels of impact was key. For example, reducing low volume surgery

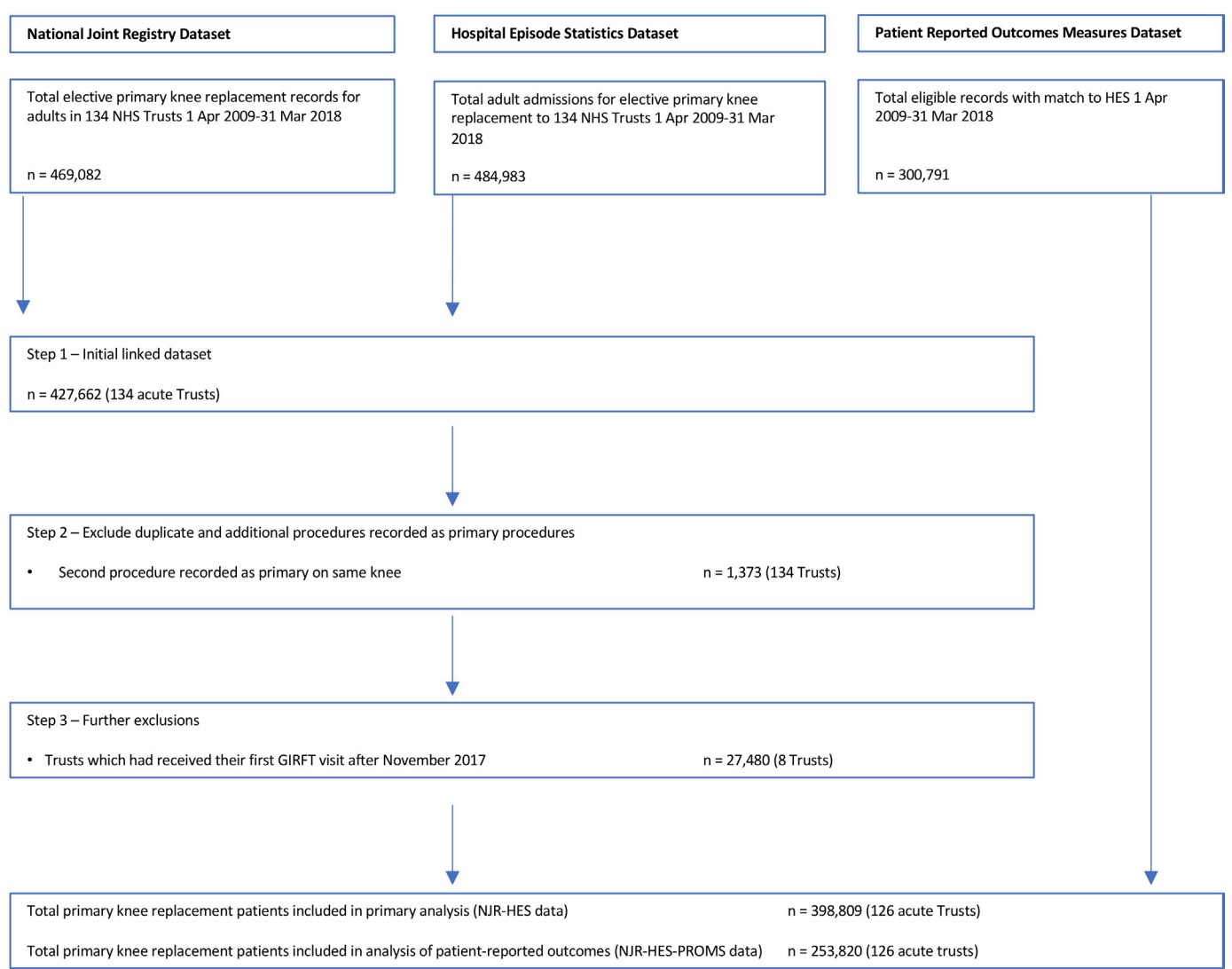

**Figure 3** Study population—total knee replacement patients. Patient-reported outcomes measures (PROMs) data collected from April 2009 for consenting patients only. GIRFT, Getting it Right First Time; HES, Hospital Episode Statistics; NHS, National Health Service; NJR, National Joint Registry.

depended on regional partnersforming networks. Similarly, increasing the number of procedures conducted per day required within-organisation negotiation to access theatre time. Visits were most successful when GIRFT aligned with Trust priorities(eg, rationalising procurement—site 3). They were less effective where GIRFT measures were not a local priority because of more immediate demands, for example, financial pressures

A range of factors, in addition to GIRFT, impacted practice. These included the concurrentroll out of other initiatives. For example, the Carter Review impacted procurement (sites 2 and 6), while reductions in arthroscopies were driven by the Payment by Results Assurance Framework (site 2).

## DISCUSSION

This is the first independent evaluation of GIRFT. We found substantial improvements in orthopaedic care during the first 6 years of the programme, notably

reductions in uncemented hip prostheses, knee arthroscopies and length of stay. However,these started before GIRFT. It was not possible to estimate the distinct contribution of the programme, because of other concurrent initiatives with common goals (eg, carter review). We also estimated the additional impact of GIRFT visits. We found a mix of positive and negative effects, generally small compared tooverall improvements and differing between the earliest and latest Trusts visited. It is important to note, though, that the eight measures we analysed in our quantitative analyses, and targeted by GIRFT, relate to direct patient care. Staff at our case study sites reported that the programme had had an impact, but the effects that they described related much more to ways of working at Trust level (eg, improved networking) rather than direct patient care.

Our mixed-method approach has enabled us to provide a comprehensive and robust understanding of GIRFT, using quantitative and qualitative data to explore the

### Hip replacements by low-volume surgeons

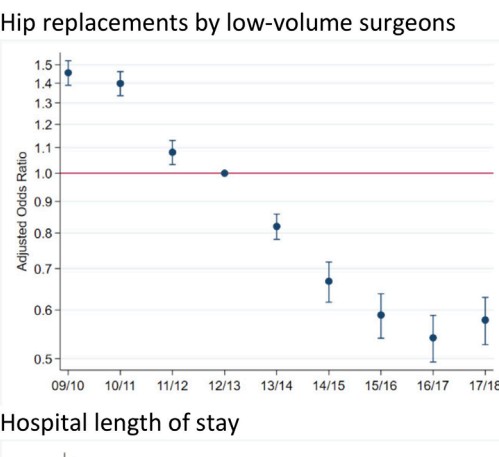

### Hospital length of stay

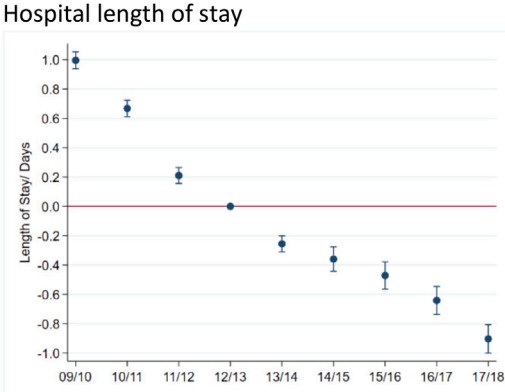

### Revision surgery on same side within 12 months

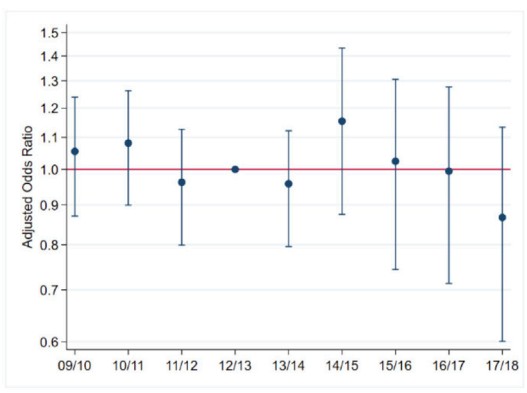

### Functional status at 6 months

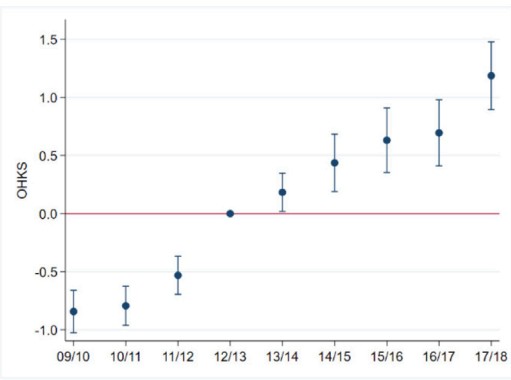

### Uncemented hip prostheses in over 65s

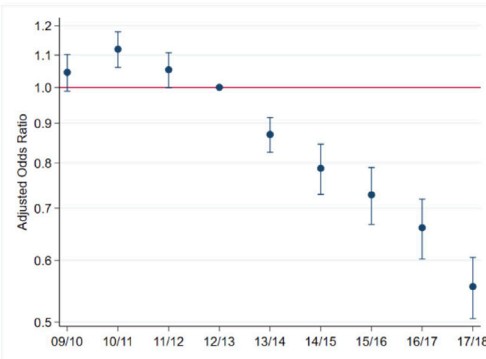

### Emergency readmission within 30 days

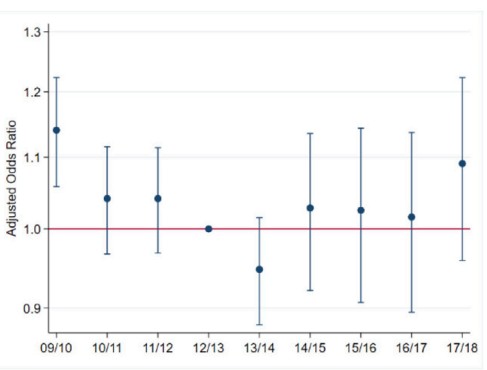

### Quality of life at 6 months

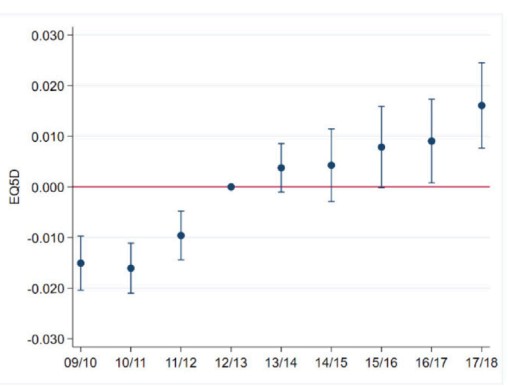

Odds ratios and mean differences reported with 95% confidence intervals from casemix-adjusted hierarchical logistic and linear regression models

**Figure 4** Trends in process and outcome measures for primary hip replacements (2009/2010–2017/2018). OHKS, Oxford Hip/Knee Score.

Knee replacements by low-volume surgeons

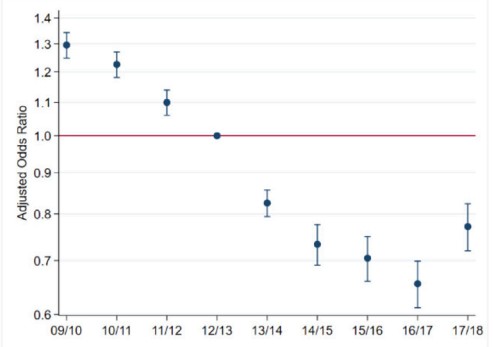

Arthroscopy on same side in previous 12 months

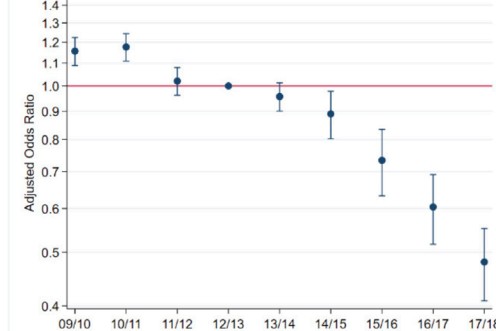

Hospital length of stay

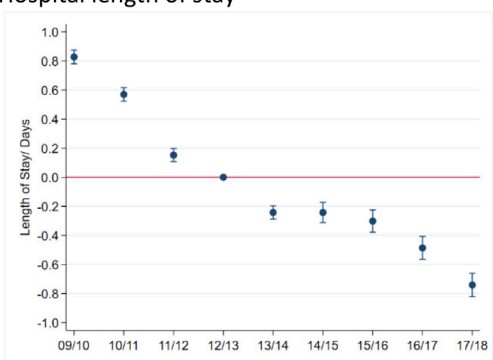

Emergency readmissions within 30 days

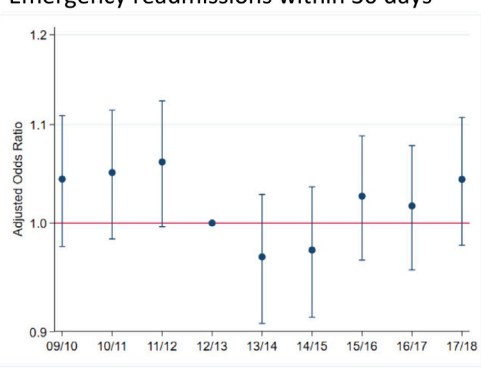

Revision surgery on same side within 12 months

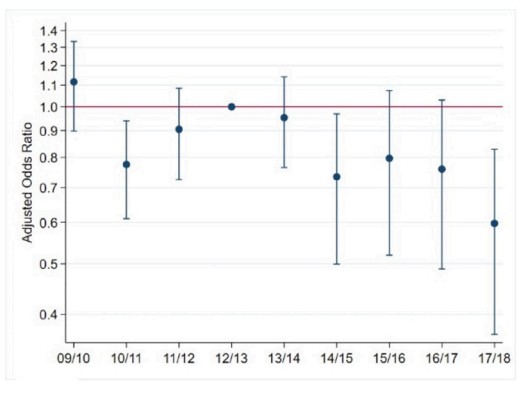

Quality of life at 6 months

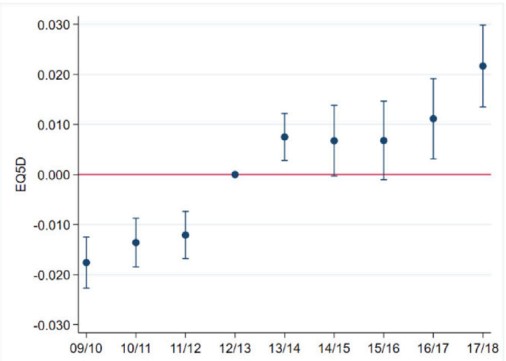

Functional status at 6 months

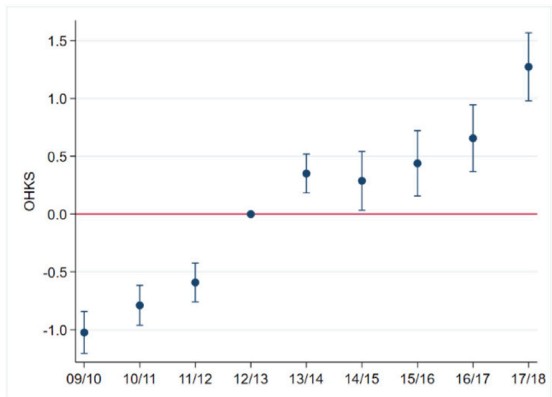

Odds ratios and mean differences reported with 95% confidence intervals from casemix-adjusted hierarchical logistic and linear regression models

**Figure 5** Trends in process and outcome measures for primary knee replacements (2009/2010–2017/2018). OHKS, Oxford Hip/Knee Score.

**Table 2** Unadjusted process and outcome measures across the study population at the beginning and end of the study period, 1 April 2009–31 March 2018

| | Beginning of study period (1 April 2009–31 March 2012) | End of study period (1 April 2016–31 March 2018) |
|---|---|---|
| Process measures | | |
| Low volume surgery (<35 per annum) (%) | | |
| Hip | 15.9 (95% CI 15.70 to 16.2) | 9.1 (95% CI 8.9 to 9.3) |
| Knee | 18.6 (95% CI 18.4 to 18.8) | 13.4 (95% 13.2 to 13.7) |
| Uncemented hip implant in >65 s (%) | 31.7 (95% CI 31.4 to 32.1) | 22.6 (95% CI 22.20 to 23.0) |
| Arthroscopy in previous 12 months (%) | 6.5 (95% CI 6.3 to 6.56) | 3.4 (95% CI 3.3 to 3.5) |
| Outcome measures | | |
| Length of stay (median and mean days) | | |
| Hip | 5 (IQR 3 to 6) 5.6 (95% CI 5.6 to 5.6) | 3 (IQR 2 to 5) 4.3 (95% CI 4.3 to 4.4) |
| Knee | 5 (IQR 3 to 6) 5.5 (95% CI 5.5 to 5.5) | 4 (IQR 3 to 5) 4.3 (95% CI 4.3 to 4.4) |
| Readmissions in 30 days (%) | | |
| Hip | 4.7 (95% CI 4.5 to 4.8) | 4.6 (95% CI 4.4 to 4.7) |
| Knee | 5.2 (95% CI 5.1 to 5.3) | 5.0 (95% CI 4.9 to 5.2) |
| Revisions in 12 months (%) | | |
| Hip | 0.77 (95% CI 0.72 to 0.82) | 0.63 (95% CI 0.58 to 0.69) |
| Knee | 0.44 (95% CI 0.40 to 0.48) | 0.37 (95% CI 0.33 to 0.41) |
| Post-operative Oxford Hip/Knee Score (mean) | | |
| Hip | 38.1 (95% CI 38.0 to 38.2) | 39.2 (95% CI 39.2 to 39.3) |
| Knee | 33.7 (95% CI 33.6 to 33.8) | 35.3 (95% CI 35.2 to 35.4) |
| Postoperative EQ5D score (mean) | | |
| Hip | 0.76 (95% CI 0.76 to 0.77) | 0.78 (95% CI 0.78 to 0.78) |
| Knee | 0.70 (95% CI 0.70 to 0.70) | 0.73 (95% CI 0.73 to 0.73) |

impact of the programme from different perspectives. Previous similar evaluations tend to focus on quantitative analyses. Our comprehensive linked dataset allowed us to examine a range of measures, as well as estimating the specific contribution of Trust visits. The only variables with missing data were ethnicity (3.8% and 2.7% of THR and TKR patients, respectively), for which we included a 'missing' category, and Index of Multiple Deprivation. We excluded the latter from the complete-case analysis, but as this represented only 0.3% of the analysis sample, it is unlikely to have had a major impact on our findings. The case study analysis provided further insights from the perspective of Trust staff. However, we measured changes at 3 months postvisit, and although some made improvements within this window, it may not have been sufficient for others. A further limitation is that we could not examine other key target measures. Procurement data were only available for 2017–2019 and could not be compared with previous years. Although litigation data are available, our previous work[24] demonstrates a significant lag from incident to resolution. Consequently, it would not be possible to determine whether changes were an impact of GIRFT or other policies (eg, Sign up to Safety). Other outcomes, such as 5-year and 10-year revisions, were beyond our time frame, as were impacts

after 2018.We also could not capture costs incurred by Trusts, because activities were not consistently tracked. The increased net cost associated with the programme is therefore a conservative estimate. Finally, our qualitative interviews took place several months after the first GIRFT visits, creating a risk of recall bias.[25]

In 2020, the GIRFT team published an internal evaluation[20] describing how the orthopaedic workstream had supported Trusts to release 'financial opportunities' of £696 million. Our findings differ for a number of reasons. First the internal evaluation was limited to descriptive analyses of national trends between 2013/2014 and 2018/2019. This is broadly consistent with our national trend analysis, although we adjusted for casemix and included data from 2009 to explore changes prior to GIRFT. In contrast, the GIRFT team attributed all trend changes to the programme. Second, they limited their economic analysis to the impact on processes and outcomes, whereas we also examined the cost of the visits and costs incurred by Trusts. Finally, our diverse case study sites facilitated cross-case comparison, to create a detailed contextual picture. The internal evaluation includes individual case studies which exemplify success. Early narrative reviews of GIRFT were published by the Kings Fund[26] and NHS Providers[27] However, these are not formal evaluations.

**Table 3** Changes in process and outcome measures after first GIRFT visit by trust cohort*

| | Early cohort 10 September 2013–2 February 2014 (n=31 Trusts) | Middle cohort 3 February 2014–6 July 2014 (n=61 Trusts) | Late cohort 7 July 2014–27 November 2017 (n=34 Trusts) |
|---|---|---|---|
| Process measures (ORs below 1 indicate improvement) | | | |
| Low volume surgery (<35 per annum) (OR) | | | |
| Hip | **0.92 (95% CI 0.87 to 0.98)** | **1.12 (95% CI 1.06 to 1.18)** | **1.11 (95% CI 1.04 to 1.19)** |
| Knee | **0.92 (95% CI 0.87 to 0.97)** | **1.13 (95% CI 1.08 to 1.19)** | **1.19 (95% CI 1.13 to 1.26)** |
| Uncemented hip implant in >65 s (OR) | 1.07 (95% CI 1.00 to 1.16) | **0.82 (95% CI 0.77 to 0.87)** | **0.73 (95% CI 0.69 to 0.78)** |
| Arthroscopy in previous 12 months (OR) | 1.09 (95% CI 0.99 to 1.20) | 1.08 (95% CI 0.99 to 1.18) | **1.19 (95% CI 1.06 to 1.33)** |
| Outcome measures (negative change in length of stay, ORs below 1, and positive changes in Oxford Hip/Knee and EQ5D scores indicate improvement) | | | |
| Length of stay (change in mean days) | | | |
| Hip | **–0.14 (95% CI -0.20 to -0.07)** | **0.09 (95% CI 0.03 to 0.14)** | **0.11 (95%CI 0.04 to 0.18)** |
| Knee | –0.01 (95% CI –0.06 to 0.04) | **0.07 (95% CI 0.03 to 0.12)** | **0.06 (95% CI 0.01 to 0.12)** |
| Readmissions within 30 days (OR) | | | |
| Hip | 1.01 (95% CI 0.92 to 1.10) | 1.07 (95% CI 0.99 to 1.16) | 1.03 (95% CI 0.94 to 1.13) |
| Knee | 1.08 (95% CI 0.99 to 1.17) | 1.03 95% CI (0.96 to 1.11) | **1.16 (95% CI 1.06 to 1.26)** |
| Revisions within 12 months (OR) | | | |
| Hip | 1.14 (95%CI 0.91 to 1.44) | 1.06 (95% 0.87 to 1.30) | 1.16 (95% CI 0.92 to 1.47) |
| Knee | 1.14 (95% CI 0.94 to 1.38) | 0.95 (95% 0.82 to 1.10) | 0.84 (95% CI 0.68 to 1.04) |
| Oxford Hip/Knee Score (change in mean score) | | | |
| Hip | **–0.12 (95% CI –0.33 to -0.09)** | –0.17 (95% CI –0.35 to 0.01) | –0.27 (95% CI –0.48 to 0.05) |
| Knee | **–0.37 (95% CI –0.58 to –0.15)** | **–0.34 (95% CI –0.53 to –0.16)** | **–0.37 (95% CI –0.59 to –0.14)** |
| EQ-5D (change in mean score) | | | |
| Hip | 0.00 (95% CI 0.00 to 0.01) | 0.00 (95% CI 0.00 to 0.01) | 0.00 (95% CI 0.00 to 0.00) |
| Knee | 0.00 (95% CI –0.01 to 0.01) | 0.00 (95% CI –0.01 to 0.01) | –0.01 (95% CI –0.01 to 0.00) |

Statistically significant values in bold.
*ORs and linear regression coefficients for postfirst GIRFT visit (reference prefirst GIRFT visit) adjusted for patient baseline characteristics and time variables.
GIRFT, Getting it Right First Time.

Our finding of improvements in processes but less clear changes in outcomes is consistent with evidence that improvement initiatives generally have greater impact on processes of care than patient outcomes.[28] Improvements observed before 2012 may be explained by GIRFT identifying existing best practice to share more widely. The additional impact of visits was mixed, with no consistent pattern across the cohorts. In some cases, performance worsened immediately afterwards. There may have been underlying differences between Trusts visited earlier and later, but visits were just one part of the programme and we are aware that other components of GIRFT, such as national reports, had impacted the care provided at case study sites. One further possible explanation is that later cohorts had made changes prior to their visit because of information they gleaned from Trusts visited earlier in the process. Although some of the Trusts were familiar with the overall recommendations being made by GIRFT, this could equally have been because they were also priorities for other national programmes being rolled out at the same time. As data collection at the case study sites took place before the quantitative analyses, and therefore, we did not know about the variations in individual process and outcome measures at the time the interviews took place, participants were not directly asked about them. It would also have been a challenge to draw firm

**Table 4** Summary of GIRFT economic impact following first GIRFT visit by trust cohort (2019/2020 GBP)

| | Early cohort 10 September 2013–2 February 2014 (n=31 Trusts) | Middle cohort 3 February 2014–6 July 2014 (n=61 Trusts) | Late cohort 7 July 2014–27 November 2017 (n=34 Trusts) | Total results for all cohorts |
|---|---|---|---|---|
| Process measures | | | | |
| Uncemented hip implant in >65 s | 87 654 | **−444 107*** | **−357 151*** | −713 604 |
| Arthroscopy in previous 12 months | 242 365 | 330 927 | **320 467*** | 893 759 |
| Outcome measures | | | | |
| Length of stay (LOS) | | | | |
| Hip | −198 320 | **5 641 544*** | **2 513 717*** | 7 956 941 |
| Knee | **1 048 798*** | **5 555 429*** | **2 333 021*** | 8 937 249 |
| Readmissions within 30 days | | | | |
| Hip | 3332 | 37 799 | 7364 | 48 495 |
| Knee | 37 325 | 21 785 | **48 943*** | 108 053 |
| Revisions within 12 months | | | | |
| Hip | 411 777 | 290 035 | 327 938 | 1 029 750 |
| Knee | 305 944 | −182 231 | −238 211 | −114 498 |
| Totals | | | | |
| Overall economic impact† | **1 938 876** | **11 251 181** | **4 956 088** | **18 146 144** |
| Savings† | −198 320 | −626 338 | −595 362 | −1 420 020 |
| Incremental cost† | 2 137 195 | 11 877 519 | 5 551 450 | 19 566 164 |
| Statistically significant economic impact‡ | 1 048 798 | 10 752 866 | 4 858 997 | 16 660 661 |
| Statistically significant economic impact excluding LOS)‡ | – | **−444 107*** | **12 259*** | **−431 848*** |

Positive values are costs; negative values are savings; statistically significant values are in bold, with reference to prefirst GIRFT visit.
*Statistically significant results at the 5% level.
†This is the sum of economic impact (saving or incremental cost) irrespective of statistically significant results.
‡This is the sum of the economic impact taking into account only statistically significant results.
GIRFT, Getting it Right First Time.

conclusions about the causes of these differences from just six sites. However, our findings reflect other literature illustrating the challenges for improvement programmes in outperforming the secular trend, including the possibility that the programmes themselves may be implicated in that trend.[29]

GIRFT is one of the largest improvement initiatives in the NHS. Our analysis demonstrates significant improvements in orthopaedic care, which began prior to GIRFT. Changes observed over the past 10 years are likely attributable to both GIRFT and other concurrent initiatives, but we cannot determine the relative contributions of

| Level of impact | Type of impact | Examples |
|---|---|---|
| **Individual practitioner** | 1. Changing clinical practice | Reduction in procedures conducted by low volume surgeons (Site 2) Informed choices about implants (e.g. uncemented hips) (Sites 1+3) |
| **Trust** | 2. Catalysing already planned changes | Expedited existing plans to improve implant selection and rationalisation of procurement (Site 3) |
| | 3. Internal data review | Greater use of local data to inform decision-making (All sites) |
| | 4. Influencing ongoing strategic conversations | Helped maintain focus on existing improvement agendas (e.g. ring fenced beds) (All) |
| | 5. Increasing collaborative working | Local: introduction of weekly, multidisciplinary team review of cases (Site 3) |
| **Regional** | | Regional: more networking with neighbouring Trusts (Sites 1+2) |

**Figure 6** Summary of impacts attributed to GIRFT orthopaedic visits at six case-study site. GIRFT, Getting it Right First Time.

these. The additional impact of visits was mixed. Given the substantial cost and expansion of the programme, ongoing monitoring and access to additional relevant data, including details of Trust activities, as well as early engagement with rigorous evaluation design (eg, stepped wedge approaches), are recommended to enhance the ability to hold GIRFT and other national improvement progammes to account.

**Author affiliations**
[1]Department of Applied Health Research, University College London, London, UK
[2]Department of Health Services Research and Policy, London School of Hygiene and Tropical Medicine, London, UK
[3]Care Policy & Evaluation Centre, London School of Economics and Political Science, London, UK
[4]Warwick Medical School, University of Warwick, Coventry, UK
[5]Department of Public Health and Primary Care, University of Cambridge, Cambridge, UK

**Acknowledgements** We thank Patricia Hallam for administrative support for the study. We thank the patients and staff of all the hospitals in England, Wales and Northern Ireland who have contributed data to the National Joint Registry. We are grateful to the Healthcare Quality Improvement Partnership (HQIP), the NJR Research Committee and staff at the NJR Centre for facilitating this work. The authors have conformed to the NJR's standard protocol for data access and publication.

**Contributors** HB was principal investigator. HB and RR initiated the research. HB, AH, EP, NJF, SM and RR designed the evaluation. SJ, AH and RG created the linked dataset. RG and AH conducted the quantitative analyses; PM provided statistical advice. SJ and EP designed the Trust survey and analysed the resultant data. EP and SM led the economic analysis. SJ and JL were responsible for qualitative data collection; FA, JL, SJ, HB and NJF analysed the qualitative data. HB, AH, EP, FA and RR drafted the manuscript and SJ, RG, JL, RM, JM, PM, NJF and SM contributed to reviewing and revision. HB, AH, EP, FA, SJ, RG, JL, RM, JM, PM, NJF, SM and RR approved the final version. HB, AH, EP, FA, SJ, RG, JL, RM, JM, PM, NJF, SM and RR had full access to all the data in the study and accept responsibility to submit for publication. HB will act as guarantor.

**Funding** This report is independent research funded by the National Institute for Health Research ARC North Thames (award/ grant number is not applicable).

**Disclaimer** The views expressed represent those of the authors and do not necessarily reflect those of the National Joint Registry Steering Committee or the Healthcare Quality Improvement Partnership (HQIP) who do not vouch for how the information is presented.

**Competing interests** All authors have completed the ICMJE uniform disclosure form at www.icmje.org/ coi_disclosure.pdf and declare: HB, AH, EP, FA, SJ, RG, JL and RR are funded in full or in part by the National Institute for Health Research ARC North Thames. RR and NJF are NIHR Senior Investigators; no financial relationships with any organisations that might have an interest in the submitted work in the previous 3 years; no other relationships or activities that could appear to have influenced the submitted work.

**Patient and public involvement** Patients and/or the public were involved in the design, or conduct, or reporting, or dissemination plans of this research. Refer to the Methods section for further details.

**Patient consent for publication** Not applicable.

**Ethics approval** This study involves human participants and was approved by NHS North West - Liverpool East Research Ethics Committee (16/NW/0654). Participants gave informed consent to participate in the study before taking part.

**Provenance and peer review** Not commissioned; externally peer reviewed.

**Data availability statement** Data are available on reasonable request. In line with the data sharing agreements between University College London and NHS Digital/ HQIP, aggregate small number suppressed outputs for the study period (1 April 2009–31 March 2018) are available on request from the corresponding author, AH ( Andrew.Hutchings@lshtm.ac.uk).

**ORCID iDs**
Helen Barratt http://orcid.org/0000-0002-1387-137X
Andrew Hutchings http://orcid.org/0000-0003-0215-9923
Elena Pizzo http://orcid.org/0000-0003-0790-7505
Fiona Aspinal http://orcid.org/0000-0003-3170-7570
Sarah Jasim http://orcid.org/0000-0003-3940-6350
Rafael Gafoor http://orcid.org/0000-0001-6865-1270
Jean Ledger http://orcid.org/0000-0003-2523-7971
Raj Mehta http://orcid.org/0000-0002-4003-530X
James Mason http://orcid.org/0000-0001-9210-4082
Peter Martin http://orcid.org/0000-0003-4638-0638
Naomi J Fulop http://orcid.org/0000-0001-5306-6140
Stephen Morris http://orcid.org/0000-0002-5828-3563
Rosalind Raine http://orcid.org/0000-0003-0904-749X

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
