## [Reviewer comments · BMJ Open]

ARTICLE DETAILS

TITLE (PROVISIONAL)	A mixed methods evaluation of the Getting it Right First Time programme in elective orthopaedic surgery in England: an analysis from the National Joint Registry and Hospital Episode Statistics
AUTHORS	Barratt, Helen; Hutchings, Andrew; Pizzo, Elena; Aspinal, Fiona; Jasim, Sarah; Gafoor, Rafael; Ledger, Jean; Mehta, Raj; Mason, James; Martin, Peter; Fulop, Naomi; Morris, Stephen; Raine, Rosalind

VERSION 1 – REVIEW

REVIEWER	Foy, Robbie University of Leeds, Leeds Institute of Health Sciences
REVIEW RETURNED	26-Nov-2021

GENERAL COMMENTS	This paper describes a mixed-methods evaluation of a national quality improvement programme, Getting it Right First Time (GIRFT), in the English NHS. GIRFT comprises specialty-specific (orthopaedics in this case), locally-tailored, comparative feedback of performance data and 'deep-dive' follow-up visits to identify how improvements can be made. The quantitative evaluation used data linkage to assess changes in recommended processes of care and patient outcomes for three years before and three years later in the programme. An analysis of national trends assessed annual changes from the start of the programme in 2012 using hierarchical regression adjusted for casemix. The study also examined the additional effects of 'deep dive' visits using before-after comparisons over three periods – the first quarter of visits, the middle half of visits, and the last quarter of visits. An economic evaluation considered direct cost savings against the costs of delivering the programme, including the costs of staff conducting visits and the opportunity costs of staff in trusts. Insufficient data were available to estimate the costs of actions taken in response to visits. The qualitative evaluation included case studies in six purposively selected trusts. Notably, this is the first independent evaluation of a GIRFT programme and, as such, is a welcome contribution (except perhaps to advocates of GIRFT). I found the paper well and concisely written. The key issue concerning the evaluation of GIRFT effects is that the study design had limited internal validity in the absence of any contemporaneous controls. The research team fully recognised this
---

	and tackled the evaluation from different angles using the most robust approaches available in the circumstances. Most outcomes appeared to be improving before GIRFT and there is little strong evidence that the programme had any impact over and above underlying trends. Furthermore, any further improvements may or may not have been attributable to other initiatives. It is also feasible, if unlikely, that GIRFT may have had protective effects on outcomes, e.g. from austerity drives over the previous decade. For the evaluation of any additional effect of the ‘deep-dive’ visits, the order of visits was not randomly decided. If so, how was the order decided? If GIRFT initially targeted poorer performing trusts, with greater scope for improvement, then greater pre-post improvements in outcomes might be expected for the earlier quarter of visits. The potential impact of contemporaneous events still hinders interpretation. For example, the research team observed reductions in procedures by low volume surgeons in the first quarter but increases in the middle half and last quarter. Did the case studies throw any light on this? Could the increases have been related to something like consultant expansion, with newer cohorts of surgeons establishing caseloads? (I’m grasping at straws there and I don’t think that this particular line of speculation needs to be covered in the manuscript!) The direct and opportunity costs of the visits more than offset the estimated savings, not accounting for any costs associated with local implementation activities that could not be identified. Is it therefore worth highlighting in the discussion that the increased net cost is a conservative estimate? The authors are very careful in considering attribution of any effects to GIRFT throughout the paper but I wondered if this sentence in the conclusion put too much emphasis on attribution to GIRFT given several uncertainties: “Our analysis demonstrates significant improvements in orthopaedic care over the past 10 years, likely attributable to both GIRFT and other concurrent initiatives.” Would it be true to say that we simply cannot tell? I liked the phrasing about enhancing the ability to hold GIRFT and other national improvement programmes to account. I suspect that the authors are pushed to keep within word limits but is there scope for being more explicit here and suggesting the need to engage with rigorous evaluation designs (e.g. stepped wedge for programmes rolling out a discrete component such as visits)? On a minor point, can the abstract explicitly state when GIRFT began?
--	--

REVIEWER	Canizares, Mayilee University Health Network, Arthritis Program
REVIEW RETURNED	24-Jan-2022

GENERAL COMMENTS	This paper evaluated the orthopaedic workstream of a national improvement programme in the NHS. The use of quantitative and qualitative methods is a strength of the study. Overall the paper is well written and easy to follow. The analyses seem mostly appropriate for the research questions of interest. Specific comments/questions are below:
--

	 • Can the authors explain a little bit more why examining the impact of the first visit to Trusts is important? Methods:  • At the beginning of the Statistical analyses sections is stated that comparisons were made before GIRFT (2009/10-2011/12) and final years of the study (2015/16-2017/18), but in table 2 the time periods are different to these ones. Please clarify. • Are the analyses presented in Table 2 unadjusted? • Table 3, what is the reference category for the ORs, before implementation? Please add explanation in the Table. • Similar comment for Table 4, what is the comparison value for the statistically significant statement? • Is the definition of 'low volume' surgeon standard or based on the study sample? • "... casemix-adjusted hierarchical logistic and linear regression..." Please explain this more. Does this mean that casemix was included in the models as a variable or analyses were stratified? • The authors conducted a complete case analysis but included a missing category for ethnicity. I think is important to expand on the amount of missing data and at the very list discuss briefly differences between missing records to those included in the analysis, as this can have an impact on the interpretation of results. This should be included on the limitations.  • Overall I think the discussion is adequate but I suggest softening the language as many of the changes observed precede the implementation of GIRFT and its is very difficult to attribute these changes to the programme.
--	--

REVIEWER	Prada, Carlos McMaster University, Orthopaedics
REVIEW RETURNED	10-Feb-2022

GENERAL COMMENTS	Thank you for submitting your valuable work to the journal. This is a mixed-method study assessing a quality improvement (QI) strategy implemented in a national health system. The purpose was to determine the impact that this QI program have had in orthopaedic surgery since its implementation. Main comments:  1. This is a well conducted mixed-method approach study. The study was planned and executed much likely to what was described in the already published research protocol and was able to assess some (but not all) of the outcomes associated with the implementation of a QI program. It is unclear to me, considering the protocol if in this manuscript they collapsed the information that was supposed to be gathered in studies 1,2 and 3 in just one study. And if so, why did the authors did not incorporate patients perspectives (study 4) in this manuscript? 2. The authors mentioned they used a semi-structured interview for the qualitative portion of the study. It would be better to have this semi-structured interview (themes/topics) as an appendix in this manuscript. In addition, to have more information regarding the development process of that interview in the methods section would be ideal. Was it piloted? Who developed it? Was it refined during the subsequent interviews? I would like to thank you again for submitting this high quality manuscript to the journal
--

VERSION 1 – AUTHOR RESPONSE

Reviewer 1 - Dr Robbie Foy, University of Leeds	
This paper describes a mixed-methods evaluation of a national quality improvement programme, Getting it Right First Time (GIRFT), in the English NHS. GIRFT comprises specialty-specific (orthopaedics in this case), locally-tailored, comparative feedback of performance data and ‘deep-dive’ follow-up visits to identify how improvements can be made. The quantitative evaluation used data linkage to assess changes in recommended processes of care and patient outcomes for three years before and three years later in the programme. An analysis of national trends assessed annual changes from the start of the programme in 2012 using hierarchical regression adjusted for casemix. The study also examined the additional effects of ‘deep dive’ visits using before-after comparisons over three periods – the first quarter of visits, the middle half of visits, and the last quarter of visits. An economic evaluation considered direct cost savings against the costs of delivering the programme, including the costs of staff conducting visits and the opportunity costs of staff in trusts. Insufficient data were available to estimate the costs of actions taken in response to visits. The qualitative evaluation included case studies in six purposively selected trusts. Notably, this is the first independent evaluation of a GIRFT programme and, as such, is a welcome contribution (except perhaps to advocates of GIRFT). I found the paper well and concisely written.	Thank you for your feedback.
The key issue concerning the evaluation of GIRFT effects is that the study design had limited internal validity in the absence of any contemporaneous controls. The research team fully recognised this and tackled the evaluation from different angles using the most robust approaches available in the circumstances. Most outcomes appeared to be improving before GIRFT and there is little strong evidence that the programme had any impact over and above underlying trends. Furthermore, any further improvements may or may not have been attributable to other initiatives. It is also feasible, if unlikely, that GIRFT may have had protective effects on outcomes, e.g. from austerity drives over the previous decade.	Thank you for your feedback.

For the evaluation of any additional effect of the 'deep-dive' visits, the order of visits was not randomly decided. If so, how was the order decided? If GIRFT initially targeted poorer performing trusts, with greater scope for improvement, then greater pre-post improvements in outcomes might be expected for the earlier quarter of visits.	GIRFT did not initially target poorer performing trusts. We have added text to the methods section (p5) which explains how the order was decided: At the start of the programme, the GIRFT team contacted all eligible Trusts, and the timetable of the first visits was based on the order in which sites responded, rather than, for example, orthopaedic performance data.
The potential impact of contemporaneous events still hinders interpretation. For example, the research team observed reductions in procedures by low volume surgeons in the first quarter but increases in the middle half and last quarter. Did the case studies throw any light on this? Could the increases have been related to something like consultant expansion, with newer cohorts of surgeons establishing caseloads? (I'm grasping at straws there and I don't think that this particular line of speculation needs to be covered in the manuscript!)	In the discussion section (p9), we have inserted additional text to explain why the case study data cannot explain variations in the impact of GIRFT between the three cohorts, including the number of procedures carried out by low volume surgeons. This now reads: The additional impact of visits was mixed, with no consistent pattern across the cohorts. In some cases, performance worsened immediately afterwards. For example, we observed reductions in procedures by low volume surgeons in the early cohort but increases in the middle and late cohorts. There may have been underlying differences between Trusts visited earlier and later, but visits were just one part of the programme and we are aware that other components of GIRFT, such as national reports, had impacted the care provided at case study sites. One further possible explanation is that later cohorts had made changes prior to their visit because of information they gleaned from Trusts visited earlier in the process. Although some of the Trusts were familiar with the overall recommendations being made by GIRFT, this could equally have been because they were also priorities for other national programmes being rolled out at the same time. As data collection at the case study sites took place before the quantitative analyses, and therefore we did not know at the time about the variations in individual process and outcome measures (e.g. procedures by low volume surgeons), participants were not directly asked about this. It would also have been a challenge to draw firm conclusions about the causes of these differences from just six sites.
The direct and opportunity costs of the visits more than offset the estimated savings, not accounting for any costs associated with local implementation activities that could not be identified. Is it therefore worth highlighting in the discussion that the increased	We have now highlighted this as you suggest, by adding text to the limitations section of the paper (p9) to say that the increased net cost is a conservative estimate.

net cost is a conservative estimate?	
The authors are very careful in considering attribution of any effects to GIRFT throughout the paper but I wondered if this sentence in the conclusion put too much emphasis on attribution to GIRFT given several uncertainties: "Our analysis demonstrates significant improvements in orthopaedic care over the past 10 years, likely attributable to both GIRFT and other concurrent initiatives." Would it be true to say that we simply cannot tell?	We agree. We have therefore amended this part of the conclusion (p10) so that it now reads: Our analysis demonstrates significant improvements in orthopaedic care, which began prior to GIRFT. Changes observed over the past 10 years are likely attributable to both GIRFT and other concurrent initiatives, but we cannot determine the relative contributions of these.
I liked the phrasing about enhancing the ability to hold GIRFT and other national improvement programmes to account. I suspect that the authors are pushed to keep within word limits but is there scope for being more explicit here and suggesting the need to engage with rigorous evaluation designs (e.g. stepped wedge for programmes rolling out a discrete component such as visits)?	We agree. We have therefore amended the conclusion (p10), so it reads: Given the substantial cost and expansion of the programme, ongoing monitoring and access to additional relevant data, including details of Trust activities, as well as early engagement with rigorous evaluation design (e.g. stepped wedge approaches), are recommended to enhance the ability to hold GIRFT and other national improvement programmes to account.
On a minor point, can the abstract explicitly state when GIRFT began?	We have amended the abstract (p2) so that it now states that GIRFT began in 2012.
Reviewer 2: Dr Mayilee Canizares, University Health Network	
This paper evaluated the orthopaedic workstream of a national improvement programme in the NHS. The use of quantitative and qualitative methods is a strength of the study. Overall the paper is well written and easy to follow. The analyses seem mostly appropriate for the research questions of interest. Specific comments/questions are below:	Thank you for your feedback.
Can the authors explain a little bit more why examining the impact of the first visit to Trusts is important?	We have added the following text to the methods section (p5): We analysed the additional impact of initiating visits using a pre-post design. First visit dates defined pre- and post- periods for Trusts. The first visit marked the start of GIRFT's local involvement with an individual Trust. Prior to this, although a Trust may have been aware of GIRFT's national work, it would not have received the intervention tailored to their individual circumstances
At the beginning of the Statistical analyses sections is stated that comparisons were made before GIRFT (2009/10-2011/12) and final years of the study (2015/16-2017/18), but in table 2 the time periods are different to these ones. Please clarify.	We apologise for the incorrect dates we initially provided. The table is incorrect with reference to 31/12/12 (This should be 31/3/12). The text is incorrect with reference to 1/4/15. (This should be 1/4/16). The table (p13) and text (p5) have both been amended accordingly.
Are the analyses presented in Table 2 unadjusted?	Yes. We have amended the table header (p13) to make this clearer.

Table 3, what is the reference category for the ORs, before implementation? Please add explanation in the Table.	The reference category here is pre-implementation, i.e. before the first GIRFT visit. The footnote has been amended to make this clear (p14).
Similar comment for Table 4, what is the comparison value for the statistically significant statement?	The reference category here is also pre-implementation. The table header has been amended to make this clear (p15).
Is the definition of 'low volume' surgeon standard or based on the study sample?	We have added text to the methods section (p4), which explains: There is no commonly accepted threshold for surgeon volume. We therefore used the thresholds ≤ 35 similar procedures or ≤ 10 unicondylar/ patella-femoral knee replacements per year based on clinical input, the literature on 'low volume' surgical thresholds in orthopaedic surgery, discussion with the GIRFT programme team and recommendations in their 2015 programme report.²
"... casemix-adjusted hierarchical logistic and linear regression..." Please explain this more. Does this mean that casemix was included in the models as a variable or analyses were stratified?	Adjustment for casemix was made by including these variables in the models. We have amended the text to make this clearer (p5). We estimated the impact in the early, middle and late groups of trusts using interactions between the group and the change in levels. We have added a sentence describing this (p5).
The authors conducted a complete case analysis but included a missing category for ethnicity. I think is important to expand on the amount of missing data and at the very list discuss briefly differences between missing records to those included in the analysis, as this can have an impact on the interpretation of results. This should be included on the limitations.	We report all missing data in Table 1 for casemix variables (p12). Ethnicity coded as missing was included in the analyses. The Index of Multiple Deprivation is the only other variable with missing values and these represent 0.3% of the analysis sample. We have added both a footnote to Table 1 and a sentence to the methods section (p5), stating that cases missing an IMD score were excluded from the complete case analysis. Given the small (0.3%) percentage of observations excluded from the analyses due to missingness we did not investigate this further. We have also added text to the limitations section (p9), explaining that the exclusion of the cases missing an IMD score is unlikely to have had a significant impact on our findings: Our comprehensive linked dataset allowed us to examine a range of measures, as well as estimating the specific contribution of Trust visits. The only variables with missing data were ethnicity (3.8% and 2.7% of THR and TKR patients respectively), for which we

	included a 'missing' category, and Index of Multiple Deprivation. We excluded the latter from the complete case analysis, but as this represented only 0.3% of the analysis sample, it is unlikely to have had a major impact on our findings.
Overall I think the discussion is adequate but I suggest softening the language as many of the changes observed precede the implementation of GIRFT and its is very difficult to attribute these changes to the programme.	We agree and have therefore amended the conclusion (p10) so that it now says: Our analysis demonstrates significant improvements in orthopaedic care, which began prior to GIRFT. Changes observed over the past 10 years are likely attributable to both GIRFT and other concurrent initiatives.
Reviewer 3: Dr. Carlos Prada, McMaster University	
Thank you for submitting your valuable work to the journal. This is a mixed-method study assessing a quality improvement (QI) strategy implemented in a national health system. The purpose was to determine the impact that this QI program have had in orthopaedic surgery since its implementation.	Thank you for your feedback.
1. This is a well conducted mixed-method approach study. The study was planned and executed much likely to what was described in the already published research protocol and was able to assess some (but not all) of the outcomes associated with the implementation of a QI program. It is unclear to me, considering the protocol if in this manuscript they collapsed the information that was supposed to be gathered in studies 1,2 and 3 in just one study. And if so, why did the authors did not incorporate patients perspectives (study 4) in this manuscript?	We have added text to the methods section (p6), which explains that our evaluation included several qualitative elements, including interviews with the GIRFT programme team and national health leaders (to understand the development of GIRFT and its evolution over time), as well as focus groups with patients (to understand their views about the content of the GIRFT programme). However, we have also noted that, as these elements did not directly contribute to our assessment of the impact of GIRFT on processes and outcomes of care, their findings will be reported elsewhere. In the results section (p8) we also highlight that our goal in this paper is to evaluate effectiveness and cost-effectiveness so we have therefore restricted our reporting to the qualitative data sources where the impact of GIRFT on patient care was explored: interviews conducted at our six case study sites.
2. The authors mentioned they used a semi-structured interview for the qualitative portion of the study. It would be better to have this semi-structured interview (themes/topics) as an appendix in this manuscript. In addition, to have more information regarding the	We agree and have now included a detailed explanation of how the topic guides were developed, together with a summary of the themes that were addressed in the interviews with providers in Appendix 2 which is in the

development process of that interview in the methods section would be ideal. Was it piloted? Who developed it? Was it refined during the subsequent interviews?	supplementary File. We have also provided a briefer description of topic guide development in the methods section (p6), so that it now says: Here, we report relevant data collected between Oct 2016-May 2019 via semi-structured interviews. The topic guide (see Appendix 2) was developed with input from the evaluation team (e.g. to incorporate questions about resource costs and implementation) and informed by scoping discussions conducted with the team delivering GIRFT to understand the programme components. It was piloted prior to the interviews, and then refined iteratively as the study progressed, to take account of emerging findings.
--	--

VERSION 2 – REVIEW

REVIEWER	Foy, Robbie University of Leeds, Leeds Institute of Health Sciences
REVIEW RETURNED	15-Mar-2022
GENERAL COMMENTS	Thank you for considering and addressing my comments
REVIEWER	Canizares, Mayilee University Health Network, Arthritis Program
REVIEW RETURNED	28-Mar-2022
GENERAL COMMENTS	The authors have adequately answered my queries.
REVIEWER	Prada, Carlos McMaster University, Orthopaedics
REVIEW RETURNED	15-Mar-2022
GENERAL COMMENTS	Thank you for submitting your revised version of this manuscript. I believe you successfully tackled all the comments flagged by the reviewers so the manuscript improved and it is ready to be accepted in its current form.